

# NALA: a Nesterov accelerated look-ahead optimizer for deep learning

Xuan Zuo[1], Hui-Yan Li[2], Shan Gao[1], Pu Zhang[3] and Wan-Ru Du[2]

[1] School of Automation, Northwestern Polytechnical University, Xi'an, Shaanxi, China
[2] China Academy of Aerospace Systems Science and Engineering, Beijing, China
[3] School of Automation and Information Engineering, Xi'an University of Technology, Xi'an, Shaanxi, China

## ABSTRACT

Adaptive gradient algorithms have been successfully used in deep learning. Previous work reveals that adaptive gradient algorithms mainly borrow the moving average idea of heavy ball acceleration to estimate the first- and second-order moments of the gradient for accelerating convergence. However, Nesterov acceleration which uses the gradient at extrapolation point can achieve a faster convergence speed than heavy ball acceleration in theory. In this article, a new optimization algorithm which combines adaptive gradient algorithm with Nesterov acceleration by using a look-ahead scheme, called NALA, is proposed for deep learning. NALA iteratively updates two sets of weights, *i.e.*, the 'fast weights' in its inner loop and the 'slow weights' in its outer loop. Concretely, NALA first updates the fast weights $k$ times using Adam optimizer in the inner loop, and then updates the slow weights once in the direction of Nesterov's Accelerated Gradient (NAG) in the outer loop. We compare NALA with several popular optimization algorithms on a range of image classification tasks on public datasets. The experimental results show that NALA can achieve faster convergence and higher accuracy than other popular optimization algorithms.

## INTRODUCTION

The remarkable success of deep learning largely owes to the advances on large scale datasets (*Russakovsky et al., 2015*), powerful computing resources, sophisticated network architectures (*He et al., 2016*) and improved optimization algorithms (*Bottou, 1991*). The training of deep neural networks (DNNs) can be cast as the optimization of a scalar parameterized loss function, which requires minimizing with respect to its parameters. Efficient optimization algorithms make it possible to train very deep artificial neural networks with large-scale datasets. Large-scale distributed optimization algorithms, which are combined with improved learning rate scheduling schemes (*Vaswani et al., 2017*), have shown impressive performance in the optimization of stochastic objectives with high-dimensional parameter spaces (*Zuo et al., 2023*).

In the last few years, a variety of optimization algorithms have been proposed to achieve the goal that accelerates the training of DNNs. Among current DNN optimizers, stochastic gradient descent (SGD) (*Robbins & Monro, 1951*) is the earliest and also the

Corresponding author
Xuan Zuo, 1069114233@qq.com

most representative stochastic optimizer, with dominant popularity for its simplicity and effectiveness. Most of the current DNN optimization algorithms are based on SGD, and improve SGD to overcome the gradient vanishing and explosion problems (*Yong et al., 2020*). These optimization algorithms for DNN can be broadly categorized into three approaches: (i) SGD (*Robbins & Monro, 1951*) and its accelerated schemes, such as Polyak heavy ball (*Polyak, 1964*) and NAG (*Nesterov, 1983*),(ii) adaptive gradient schemes, such as AdaGrad (*Duchi, Hazan & Singer, 2011*), RMSProp (*Tieleman & Hinton, 2012*) and Adadelta (*Zeiler, 2012*), (iii) mixed schemes combining the advantages of adaptive gradient schemes and accelerated schemes, such as Adam (*Kingma & Ba, 2015*), AdamW (*Loshchilov & Hutter, 2019*), Adan (*Xie et al., 2022*), AdaXod (*Liu & Li, 2023*), Nadam (*Dozat, 2016*), AMSGrad (*Reddi, Kale & Kumar, 2018*), RAdam (*Liu et al., 2020*), Ranger21 (*Wright & Demeure, 2021*) and AdamP (*Heo et al., 2021*).

Although optimization algorithms with adaptive gradient can adjust the learning rate according to the geometry curvature of the loss objective, previous work has revealed that these algorithms mainly borrow the moving average idea from heavy ball acceleration to estimate the first- and second-order moments of the gradient in order to accelerate convergence. Furthermore, Nesterov acceleration can achieve a faster convergence speed than heavy ball acceleration, since it uses the gradient at extrapolation point and sees a slight 'future' (*Xie et al., 2022*; *Nesterov, 1983*). That inspires us to combine adaptive gradient algorithms with Nesterov acceleration.

In this work, a new optimization algorithm, named Nesterov Accelerated Look-Ahead (NALA), is proposed to combine the advantages of two popular optimization algorithms: NAG, which is superior to heavy ball acceleration for conventionally difficult optimization problems (*Sutskever et al., 2013*), and Adam, which works well with sparse gradients and non-stationary objectives (*Kingma & Ba, 2015*). Moreover, the trick of maintaining two sets of weights which is used in Lookahead (*Zhang et al., 2019*) for choosing a good search direction is also applied in NALA. This trick improves learning in high curvature directions, reduces variance, and makes the optimization algorithm converge rapidly in practice (*Zhang et al., 2019*). NALA first updates the 'fast weights' $k$ times using Adam optimizer in its inner loop, and then updates the 'slow weights' once in the direction of Nesterov's accelerated gradient in its outer loop. After the slow weights update, the fast weights are reset to the current slow weights value. Since the anticipatory updates in Nesterov acceleration can prevent from going too fast and lead to the increased responsiveness, the slow weights updating in the direction of NAG can avoid the missing of the global optimum (*Lin et al., 2020*). Portions of this text were previously published as part of a preprint (*Zuo et al., 2023*).

'Related Work' provides a review of the related work. 'Nesterov Accelerated Look-Ahead Algorithm' gives the details of the proposed algorithm and its update rule. In 'Experiments', the convergence and the accuracy rate of NALA are evaluated on a range of image classification tasks on the CIFAR10, CIFAR100 (both collected from the 80 Million tiny images dataset which was withdrawn from use in 2000, https://groups.csail.mit.edu/vision/TinyImages/) and Fashion-MNIST (*Xiao, Rasul & Vollgraf, 2017*) datasets. 'Robustness to the Hyperparameters' explores the robustness

of NALA to its hyperparameters by fixing the inner optimizer and evaluating runs with varied synchronization period, decay factor and step size of slow weights. The results of our experiments show that NALA performs better than other popular optimization algorithms on the image classification models in most cases, and it is robust to a wide range of hyperparameter settings.

# RELATED WORK

This work is inspired by recent advances in improving adaptive gradient algorithms with Nesterov momentum (*Dozat, 2016*; *Li, Li & Zhang, 2021*; *Chen et al., 2022*; *Xie et al., 2022*) and the idea of parameter averaging (*Anderson, 1965*; *Nichol, Achiam & Schulman, 2018*; *Izmailov et al., 2018*; *Zhang et al., 2019*). While previous work has demonstrated the advantage of combining adaptive gradient algorithms with Nesterov momentum, incorporating Nesterov momentum into averaging weights method has not been carefully studied. The most related work to ours is Lookahead (*Zhang et al., 2019*), which performs parameter averaging to achieve faster convergence. There are a few important differences between Lookahead and NALA: Lookahead generates its parameter updates using the moving averages of its fast weights and slow weights, whereas NALA generates parameter updates by applying the Nesterov accelerated gradient of the moving averages over its weights. This section briefly reviews the related work from two aspects, *i.e.,* adaptive gradient algorithms with Nesterov momentum, and parameter averaging methods.

## Adaptive gradient algorithms with Nesterov momentum

The Nadam algorithm (*Dozat, 2016*) simplifies Nesterov acceleration to estimating the first moment of gradient in Adam. Although its acceleration does not use any gradient from the extrapolation points, the improvement of Nadam over Adam is fairly dramatic in most cases (*Dozat, 2016*). A similar algorithm is Adan (*Xie et al., 2022*), which adopts a new Nesterov momentum estimation (NME) method to estimate the first- and second-order moments of the gradients in Adam. Adan avoids the extra computation and memory overhead of computing gradient at the extrapolation point, and speedup the training of DNNs effectively (*Xie et al., 2022*). Nesterov momentum is also used for improving the rapidly promoted distributed adaptive gradient descent optimization algorithm. NDADAM (*Li, Li & Zhang, 2021*) algorithm incorporates Nesterov's momentum into distributed adaptive gradient method for online optimization. The experimental results show that the convergent speed of NDADAM has been greatly improved. NAI-FGM (*Chen et al., 2022*) is a gradient-based attack algorithm, which applies Nesterov momentum and Adam to iterative attacks to improve its transferability. NAI-FGM can not only effectively avoid local optimum, but also adaptively adjust the attack step size to reach the global optimum fast. In contrast to these approaches, which combine the advantages of NAG and Adam optimization algorithm, NALA additionally performs parameter averaging so as to take advantage of the geometry of loss surfaces to improve convergence.

## Parameter averaging methods

The parameter averaging scheme, which focuses on averaging the weights of different neural networks, have been used in natural language processing (*Jean et al., 2014*; *Merity,*

*Keskar & Socher, 2017*) and generative adversarial networks (*Yazici et al., 2018*). Anderson acceleration (*Anderson, 1965*), an algorithm of iterative procedures for nonlinear integral equations, keeps track of all iterates within an inner loop and then computes some linear combinations which extrapolate the iterates towards their fixed point. The Reptile (*Nichol, Achiam & Schulman, 2018*) algorithm, a first-order gradient-based meta-learning method, also uses an outer and inner loop during optimization. Reptile works by repeatedly sampling a task in its outer loop, training on it within the inner loop, and moving the initialization towards the trained weights on that task. Stochastic Weight Averaging (SWA) (*Izmailov et al., 2018*) is an algorithm employing the average of SGD weights with a cyclical or constant learning rate, which averages the weights of different neural networks obtained during training. SWA also leads to a better understanding of the geometry of their loss surface. Lookahead (*Zhang et al., 2019*) is a simple version of Anderson acceleration wherein only the first and last iterates are used. It avoids the challenges in the form of additional memory overhead as the number of inner-loop steps increases and finding the best linear combination of iterates. Moreover, Lookahead can combine parameter averaging with any standard optimizer. Ranger21 (*Wright & Demeure, 2021*) is a mix of several current optimization techniques which also absorbs the parameter averaging scheme used in Lookahead. Ranger21 combines AdamW (*Loshchilov & Hutter, 2019*) with eight optimizer components, and experimentally provides consistent improvements over AdamW. Our NALA algorithm, which is closely related to Lookahead, adds a Nesterov momentum on top of the Lookahead to accelerate convergence speed.

## NESTEROV ACCELERATED LOOK-AHEAD ALGORITHM

Accelerated gradient schemes were first proposed by *Polyak (1964)*. This well-known technique is called heavy ball because its idea comes from a heavier ball which intuitively bounces less and moves faster through regions of low curvature than a lighter ball due to momentum. After that, *Nesterov (1983)* demonstrated a modification to gradient descent that could obtain optimal performance for the algorithms applied to minimize smooth convex functions (*Brendan & Emmanuel, 2015*). Like heavy ball, Nesterov's Accelerated Gradient (NAG) is a first-order optimization method with better convergence rate guarantee than gradient descent in certain situations. Moreover, it has been demonstrated that NAG is in general superior to heavy ball (*Sutskever et al., 2013*). The NAG algorithm can be written as follows (*Nesterov, 1983*):

$$y_{t+1} = (1 + \mu_t)\theta_t - \mu_t\theta_{t-1},$$
$$\theta_{t+1} = y_{t+1} - \alpha_t J'(y_{t+1}), \tag{1}$$

where $\theta$ is the parameter of the objective function $J$, and $\mu_t$ is a decay factor of previous parameters at timestep $t$. NAG computes the gradient of $J$ at an extrapolation point with parameter $y_{t+1}$, which represents the moving average of previous parameters $\theta_t$ and $\theta_{t-1}$, then updates the parameter using a learning step size $\alpha_t$. As shown in Eq. (1), NAG smooths the previous two parameter values and takes a gradient descent step from the smoothed value $y_{t+1}$. *Sutskever et al. (2013)* rewrites NAG as an improved momentum method, which

can be expressed as:

$$v_{t+1} = \mu_t v_t - \alpha_t J'(\theta_t + \mu_t v_t),$$
$$\theta_{t+1} = \theta_t + v_{t+1}, \tag{2}$$

where $v_{t+1}$ is the Nesterov momenum at timestep $t$, and $\mu_t$ is the parameter of this momenum. Equation (2) reveals the relation of NAG to the Polyak heavy ball method. Compared with the heavy ball method, NAG can prevent the gradient descent from going too fast and lead to increased responsiveness, so as to avoid missing the global optimum (*Lin et al., 2019*).

Motivated by NAG, this work focuses on how to incorporate Nesterov momentum into Lookahead. Lookahead chooses a search direction by looking ahead at the sequence of 'fast weights' generated by its inner loop optimizer, and it is orthogonal to previous optimization algorithms and robust to changes in the inner loop optimizer (*Zhang et al., 2019*). Therefore, any standard optimizers can be used as the inner loop optimizer in Lookahead.

The proposed optimization algorithm, NALA, adopts a modified look-ahead scheme which incorporates Nesterov momentum into Lookahead. Like the vanilla Lookahead, NALA maintains two sets of weights (*i.e.,* fast weights in the inner loop, slow weights in the outer loop). Moreover, NALA can also combine with another standard optimizer in its inner loop. For the optimization of convex function, NALA theoretically achieves a faster convergence speed than Lookahead, as it sees a slight future at the extrapolation point by using Nesterov's momentum.

The algorithm details of NALA are shown in Algorithm 1, wherein $\theta$ denotes the fast weights for inner loop, $\phi$ denotes the slow weights for outer loop with the step size $\alpha$, and $\mu$ denotes the decay factor ($\mu < 0$). One of the good default settings for the image classification tasks in this work is $\alpha = 0.001$, $\mu = -0.5$. The synchronization period $k$ of the fast and slow weights is set to 5 in the image classification tasks below. And in 'Robustness to the Hyperparameters', it will be proved that the performance of NALA is robust to different settings of $k$. The implicit function $A$ denotes the inner loop optimizer.

Since adaptive gradient algorithms can adaptively adjust the learning rate to solve the problems that may be caused by the fixed step size, we prefer to exploit an adaptive gradient algorithm as the inner loop optimizer $A$ for our NALA. As is widely known, Adam and its variants are among the most commonly employed adaptive optimizers in deep learning (*Wright & Demeure, 2021*). In our NALA, Adam is employed as the inner loop optimizer $A$ to generate the sequence of fast weights, as it works well with sparse gradients and non-stationary objectives (*Kingma & Ba, 2015*). The algorithm details of Adam are given as *Kingma & Ba (2015)*:

$$g_t \leftarrow \nabla_{\theta_{t-1}} f(\theta_{t-1}),$$
$$m_t \leftarrow \beta_1 m_{t-1} + (1 - \beta_1)g_t,$$
$$\hat{m}_t \leftarrow \frac{m_t}{1 - \beta_1^t},$$
$$v_t \leftarrow \beta_2 v_{t-1} + (1 - \beta_2)g_t^2,$$

---

**Algorithm 1** NALA Optimizer:

---

**Require:** Initial parameters $\phi_0$, objective function $J$

**Require:** Synchronization period $k$, slow weights step size $\alpha$, decay factor $\mu$, optimizer $A$

**for** $t = 1, 2, \ldots,$ **do**

Synchronize parameters $\theta_{t,0} \leftarrow \phi_{t-1}$

**for** $i = 0, 1, 2, \ldots, k-1$ **do**

sample mini-batch of data $d \sim D$

$\theta_{t,i+1} = \theta_{t,i} + A\left(J, \theta_{t,i}, d_D\right)$

**end for**

Perform outer update

$\begin{cases} y_t = (1+\mu)\theta_{t,k} - \mu \cdot \phi_{t-1} \\ \phi_t = y_t - \alpha \cdot \nabla J(y_t) \end{cases}$

**end for**

**return** parameters $\phi$

---

$$\hat{v}_t \leftarrow \frac{v_t}{1 - \beta_2^t},$$

$$\theta_t \leftarrow \theta_{t-1} - \eta \frac{\hat{m}_t}{\sqrt{\hat{v}_t} + \epsilon}. \tag{3}$$

where $g_t$ is the gradient of the objective founction $f$ with parameters $\theta$ at timestep $t$. The first and second moment estimates of $g_t$ are denoted as $m_t$ and $v_t$ with exponential decay rates $\beta_1$ and $\beta_2$ respectively, and the bias-corrected first and second moment estimate are denoted as $\hat{m}_t$ and $\hat{v}_t$. In the last line of Eq. (3), $\epsilon$ is an extremely small positive constant. Adam combines RMSProp with classical momentum (*Dozat, 2016*), and replaces the estimated gradient $g_t$ with a moving average $m_t$ of all previous gradient $g_t$ based on RMSProp. It adjusts the learning rate for each step gradient according to the current geometry curvature of the loss objective, thus offers faster convergence speed than SGD across most DNN models (*Xie et al., 2022*).

NALA maintains a set of fast weights $\theta$ and another set of slow weights $\phi$, which get synchronized every $k$ updates. The fast weights are updated by applying the inner loop optimizer $A$ to the mini-batch training examples $d$, which are sampled from the dataset $D$. The trajectory of the fast weights $\theta$ in the inner loop is given by:

$$\theta_{t,i+1} = \theta_{t,i} + A\left(J, \theta_{t,i}, d_D\right), \tag{4}$$

where $t$ denotes the timestep of the outer loop, and $i$ denotes the timestep of the inner loop. After $k$ inner optimizer updates by using the optimizer $A$, the slow weights are updated in the direction of NAG at the extrapolation point derived from exponentially-decayed moving averages of the fast and slow weights. The trajectory of the slow weights $\phi_t$ can be characterized as an exponential moving average of the final fast weights in each inner loop $\theta_{t,k}$ and the gradient at each extrapolation point $\nabla J(y_t)$:

$$\phi_t = y_t - \alpha \cdot \nabla J(y_t)$$

$$= (1+\mu)\theta_{t,k} - \mu \cdot \phi_{t-1} - \alpha \cdot \nabla J(y_t)$$
$$= (1+\mu)\left[\theta_{t,k} + |\mu|\theta_{t-1,k} + |\mu|^2\theta_{t-2,k} + \cdots + |\mu|^{t-1}\theta_{1,k}\right] + |\mu|^t \phi_0 -$$
$$\alpha\left[\nabla J(y_t) + |\mu|\nabla J(y_{t-1}) + |\mu|^2\nabla J(y_{t-2}) + \cdots + |\mu|^{t-1}\nabla J(y_1)\right]. \tag{5}$$

According to Eq. (5), in each inner loop, only the last step of the fast weights $\theta_{t,i}$ has a direct impact on the trajectory of the slow weights. After the slow weights update, the fast weights are reset to current slow weights value.

Figure 1 illustrates the trajectories of the fast weights in the inner loop and the slow weights in the outer loop during the running of algorithm. While the fast weights explore around the minima of the loss surface, the slow weights look ahead at the extrapolation point and are then updated in the direction of NAG. Therefore, the proposed algorithm can update parameters of models to be optimized along a shortcut.

*Martens (2014)* has demonstrated that 'an exponentially-decayed moving average typically works much better in practice'. Intuitively, the combination of fast weights and slow weights can improve learning in high curvature directions, reduces oscillation, and enables this algorithm to converge rapidly (*Zhang et al., 2019*). Theoretically, oscillation often occurs in the high curvature direction, while the fast weights updates make rapid progress along the low curvature direction. Moreover, the slow weights can help smooth out the oscillation through the parameter averaging.

We evaluate the computational complexity of the proposed NALA algorithm. As NALA maintains a single additional copy of the learnable parameters of the trained model, the number of operations is $\mathcal{O}(\frac{k+1}{k})$ times that of its inner optimizer. Compared with second order methods which need to solve the intractable Hessian matrix, the computation and memory cost of this additional copy is acceptable and negligible.

## EXPERIMENTS

To evaluate the performance of our NALA algorithm, we train three classical convolutional neural networks (CNN) models with NALA and several popular optimizers for image classification on the famous public datasets, *i.e.*, CIFAR-10, CIFAR-100 (both collected from the 80 Million tiny images dataset which was withdrawn from use in 2000, https://groups.csail.mit.edu/vision/TinyImages/) and Fashion-MNIST (*Xiao, Rasul & Vollgraf, 2017*).

### Datasets

The CIFAR-10/CIFAR-100 dataset for classification tasks consists of 60,000 $32 \times 32$ color images in 10/100 classes. Each class has 6,000 images in CIFAR-10 and 600 images in CIFAR-100. The classes are completely mutually exclusive, and there is no overlap between different classes. For both CIFAR-10 and CIFAR-100, the 60,000 color images are split into a training set with 50,000 images and a test set with 10,000 images. Fashion-MNIST is a dataset comprising of $28 \times 28$ grayscale images of 70,000 fashion products from 10 categories, with 7,000 images per category. The training set of Fashion-MNIST has 60,000 images, and the test set of it has 10,000 images.

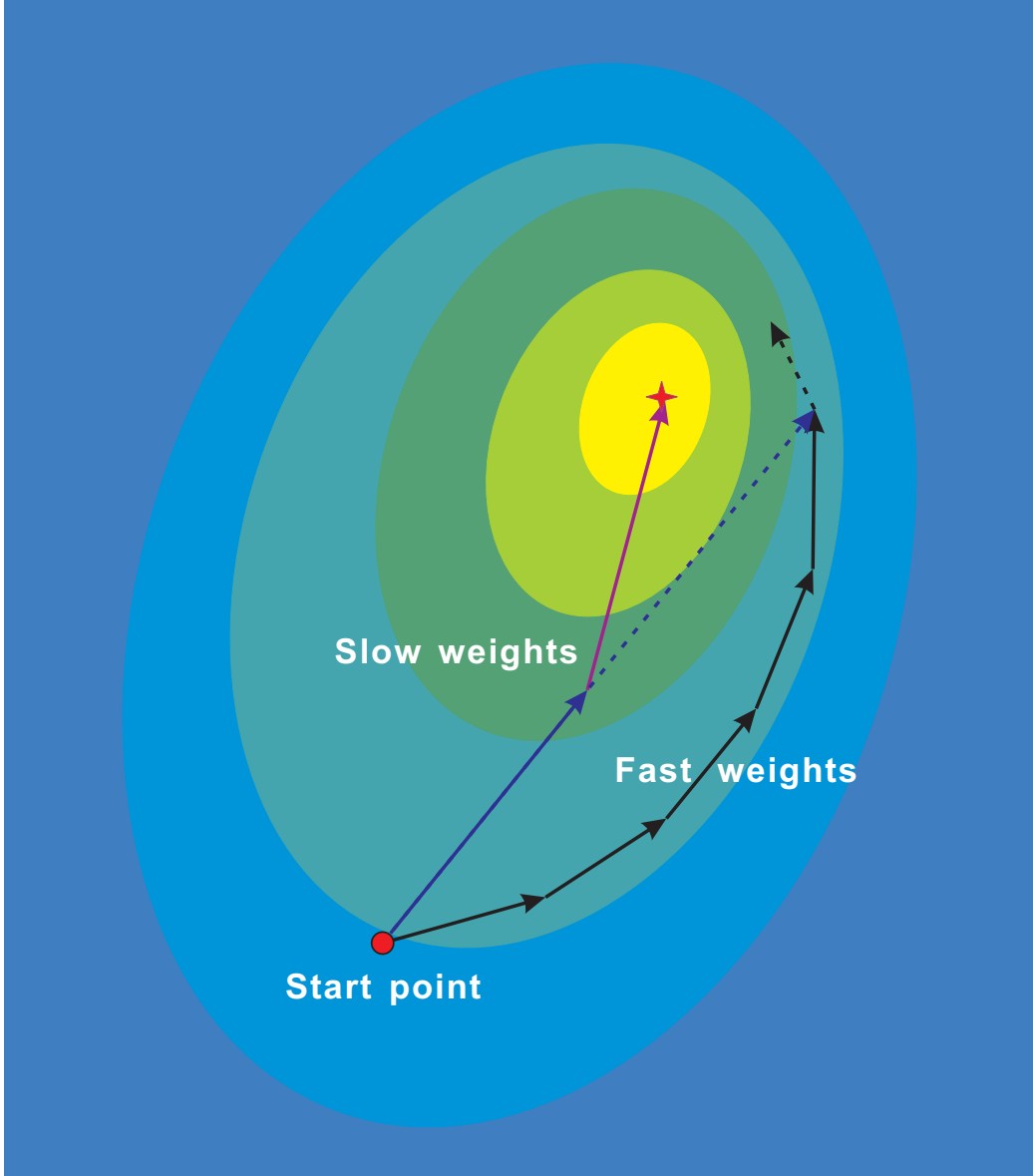

**Figure 1** **NALA trajectories of the fast weights and slow weights on loss surface.** The fast inner-loop weights explore along the black solid path. The slow outer-loop weights first go along the dark blue solid arrow to the extrapolation point, and then updates in the direction of NAG (purple solid arrow). The dark blue dashed arrow denotes the direction of the classical momentum.

In this work, three classcial CNN architectures (*i.e.,* LeNet-5, AlexNet and ResNet-18) are trained on the CIFAR-10, CIFAR-100 or Fashion-MNIST datasets with 2 random seeds and batches of 256 images. All the experiments are conducted for 230 epochs, and the learning rate is decayed every 20 epochs by an exponential decay factor of 0.5. A series of data augmentation techniques, including random crop, horizontal flipping, random brightness and random contrast, have been utilized to facilitate the learning of the three

CNN models on these classification tasks. The experimental data of these classification tasks are recorded at every 100 timesteps throughout the training process.

## Experiments on LeNet-5

To compare our NALA algorithm with other popular algorithms, this work implements five different optimization algorithms, *i.e.,* NALA, NAG, Lookahead, Adam and SGD, to train the LeNet-5 architecture on CIFAR-10 and CIFAR-100 datasets respectively. LeNet-5 (*LeCun et al., 1998*) is one among the earliest CNNs which promotes the event of deep learning. The LeNet-5 architecture consists of two sets of convolutional and average pooling layers, followed by a flattening convolutional layer, then two fully-connected layers and finally a softmax classifier.

This work uses the standard deterministic cross-entropy objective function to train LeNet-5 models with the above five optimization algorithms and shows the learning curves in Fig. 2. Since the default initial learning rate of popular optimizers is empirically effective for most optimization problems, the initial learning rate of NALA, Lookahead, and Adam is set to 0.001, while the rate for NAG and SGD is set to 0.1. The momentum parameter of NAG is empirically set to 0.1 in these experiments. For both NALA and Lookahead, the synchronization period of the weights of inner and outer loops is set to 5. The loss curves during training on CIFAR-10 and CIFAR-100 are shown in Figs. 2A and 2B, and the top-1 accuracy curves on CIFAR-10 and CIFAR-100 are shown in Figs. 2C and 2D.

As shown in Fig. 2, NALA exhibits comparable performance to Adam, and both of them outperform NAG, Lookahead and SGD on CIFAR-10. On CIFAR-100, the two algorithms also achieve significantly faster convergence and higher accuracy than Lookahead and SGD, while they have a slight advantage over NAG. It can be found that, during the early stage of training, NALA and Adam show a faster learning speed than the other algorithms. Furthermore, NALA converge to lower training loss and higher top-1 accuracy than the other algorithms at the end of training; see Table 1.

The number of timesteps the five optimization algorithms require to achieve 70% top-1 accuracy and 90% or 50% top-1 accuracy are given in Table 2. As shown in Table 2 and Fig. 2, for the CIFAR-10 and CIFAR-100 classification tasks on LeNet-5 architecture, SGD and Lookahead take much longer to converge, and they are unable to match the final performance of the other three optimizers. In contrast to the other optimization algorithms, NALA achieves a faster learning speed and higher top-1 accuracy on each image classification task.

## Experiments on AlexNet

AlexNet (*Krizhevsky, Sutskever & Hinton, 2012*) won the ImageNet Large Scale Visual Recognition Challenge (ILSVRC) 2012 by a large margin. It is considered to be the first modern CNN which uses GPU to boost performance. AlexNet represents a significant evolutionary improvement over LeNet-5, yet there are also notable differences between the two architectures. Concretely, AlexNet is much deeper than LeNet-5, and it consists of eight layers: five convolutional layers, two fully connected hidden layers, and one fully connected output layer. In addition, AlexNet changes the sigmoid activation function

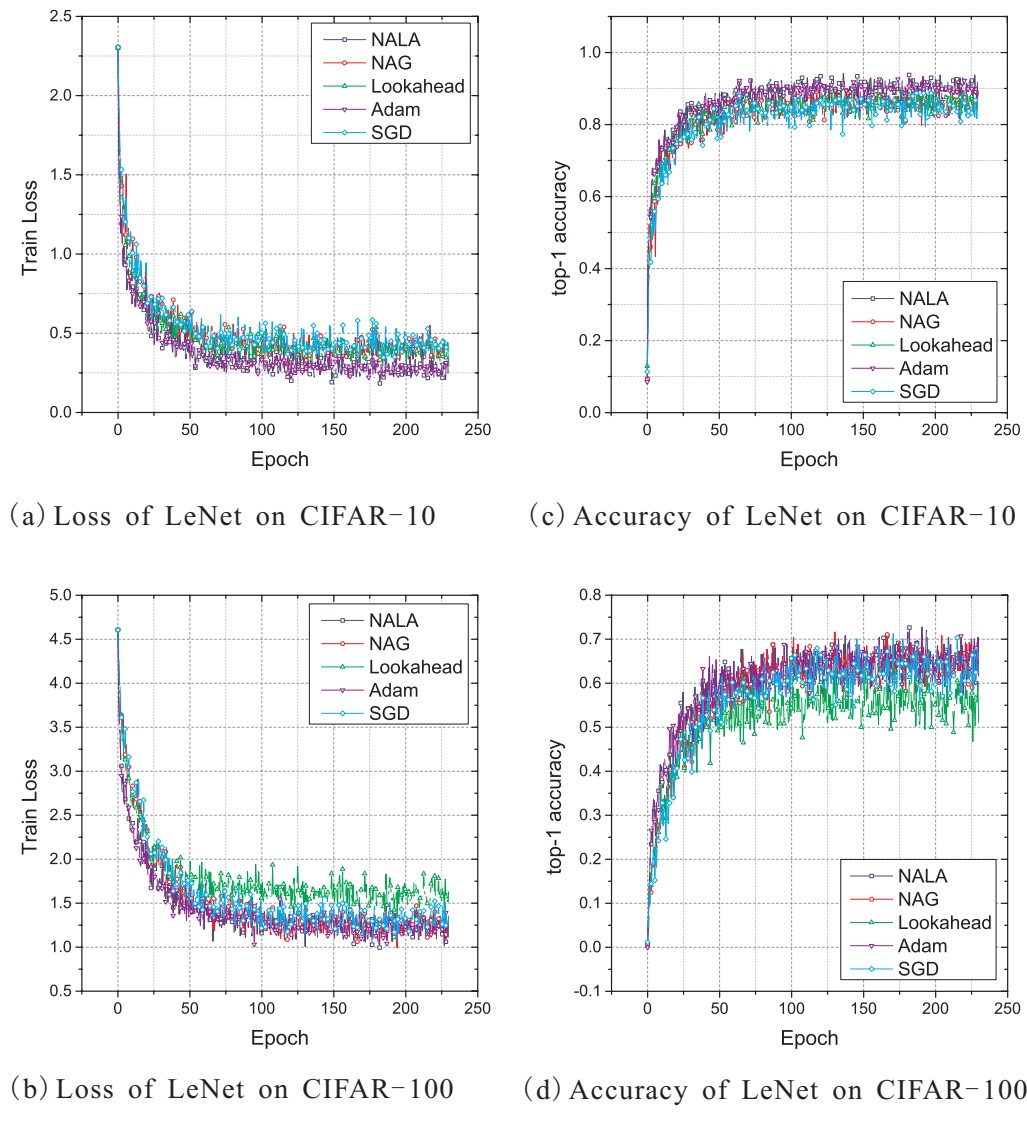

(a) Loss of LeNet on CIFAR−10

(c) Accuracy of LeNet on CIFAR−10

(b) Loss of LeNet on CIFAR−100

(d) Accuracy of LeNet on CIFAR−100

**Figure 2** (A–D) Train loss and top-1 accuracy of LeNet-5 trained by five different optimizers on CIFAR-10 and CIFAR-100.

to a simpler Rectified Linear Unit (ReLU) activation function. In this work, AlexNet is trained by using NALA, Lookahead and Adam respectively for image classification on the CIFAR-10 and CIFAR-100 datasets.

Similarly to the experiments on LeNet-5 above, the experiments on AlexNet are conducted with the cross-entropy loss. The default setting of the initial learning rate of the standard Adam optimizer, which is also a good setting for NALA and Lookahead, is set to 0.001 and applied to implement the three optimization algorithms. And the synchronization period $k$ is set to 5 for both NALA and Lookahead. Additionally, the dropout stochastic regularization (*Hinton et al., 2012*) is applied into the two fully connected hidden layers to prevent over-fitting with probability of 0.5. The training curves of loss value on CIFAR-10

Table 1 The best records of LeNet-5 train loss and top-1 accuracy on CIFAR-10 and CIFAR-100 during the 230 epochs training.

| LeNet-5 | CIFAR-10 | | CIFAR-100 | |
|---|---|---|---|---|
| | Loss value | Top-1 accuracy | Loss value | Top-1 accuracy |
| NALA | 0.182860419 | 0.94531250 | 0.961694419 | 0.73828125 |
| NAG | 0.296762913 | 0.90625000 | 1.071134210 | 0.72265625 |
| Lookahead | 0.270459294 | 0.92968750 | 1.263267040 | 0.66015625 |
| Adam | 0.201464713 | 0.94140625 | 0.986573994 | 0.72265625 |
| SGD | 0.327933371 | 0.89453125 | 1.057244420 | 0.71875000 |

Table 2 The number of timesteps these optimization algorithms require to achieve 70% top-1 accuracy and 90% or 50% top-1 accuracy during the 230 epochs training.

| LeNet-5 | CIFAR-10 | | CIFAR-100 | |
|---|---|---|---|---|
| | 70% accuracy | 90% accuracy | 50% accuracy | 70% accuracy |
| NALA | 12,00 | 12,100 | 3,200 | 25,400 |
| NAG | 1,900 | 25,700 | 4,800 | 25,400 |
| Lookahead | 2,200 | 16,500 | 6,100 | – |
| Adam | 1,00 | 12,500 | 3,500 | 27,400 |
| SGD | 1900 | – | 5200 | 28400 |

and CIFAR-100 are shown in Figs. 3A and 3B, and the curves of top-1 accuracy rate are shown in Figs. 3C and 3D.

As shown in Figs. 3A and 3C, NALA achieves slightly better train loss and top-1 accuracy than Lookahead on CIFAR-10, and both NALA and Lookahead exhibit a significant advantage over Adam. In the CIFAR-100 experiment, NALA also outperforms Adam and achieves similar performance with Lookahead, as shown in Figs. 3B and 3D. The clear advantage of NALA and Lookahead in optimizing the AlexNet model is perhaps due to the fact that the parameter averaging of the fast and slow weights smooths out the oscillation in high curvature directions, thus pushing the optimization towards an area with a lower loss value. Table 3 gives the lowest loss value and the highest top-1 accuracy rate achieved by the three optimizers during the 230 epochs training. Table 4 gives the number of timesteps the three optimization algorithms require to achieve 60% top-1 accuracy and 80% or 50% top-1 accuracy during training.

As shown in Tables 3 and 4, NALA exhibits comparable performance to Lookahead, and the two algorithms converge to higher top-1 accuracy than Adam with faster learning speeds on both the CIFAR-10 and CIFAR-100 datasets. These demonstrate the advantage of the parameter averaging method in optimizing the weights of DNNs.

## Experiments on ResNet-18

Residual Networks (ResNets) learn residual functions with reference to the layer inputs, rather than learning unreferenced functions. Instead of hoping that each few stacked layers directly fit a desired underlying mapping, ResNets let these layers fit a residual mapping. Resnet models have 5 different versions, which contain 18, 34, 50, 101 and 152

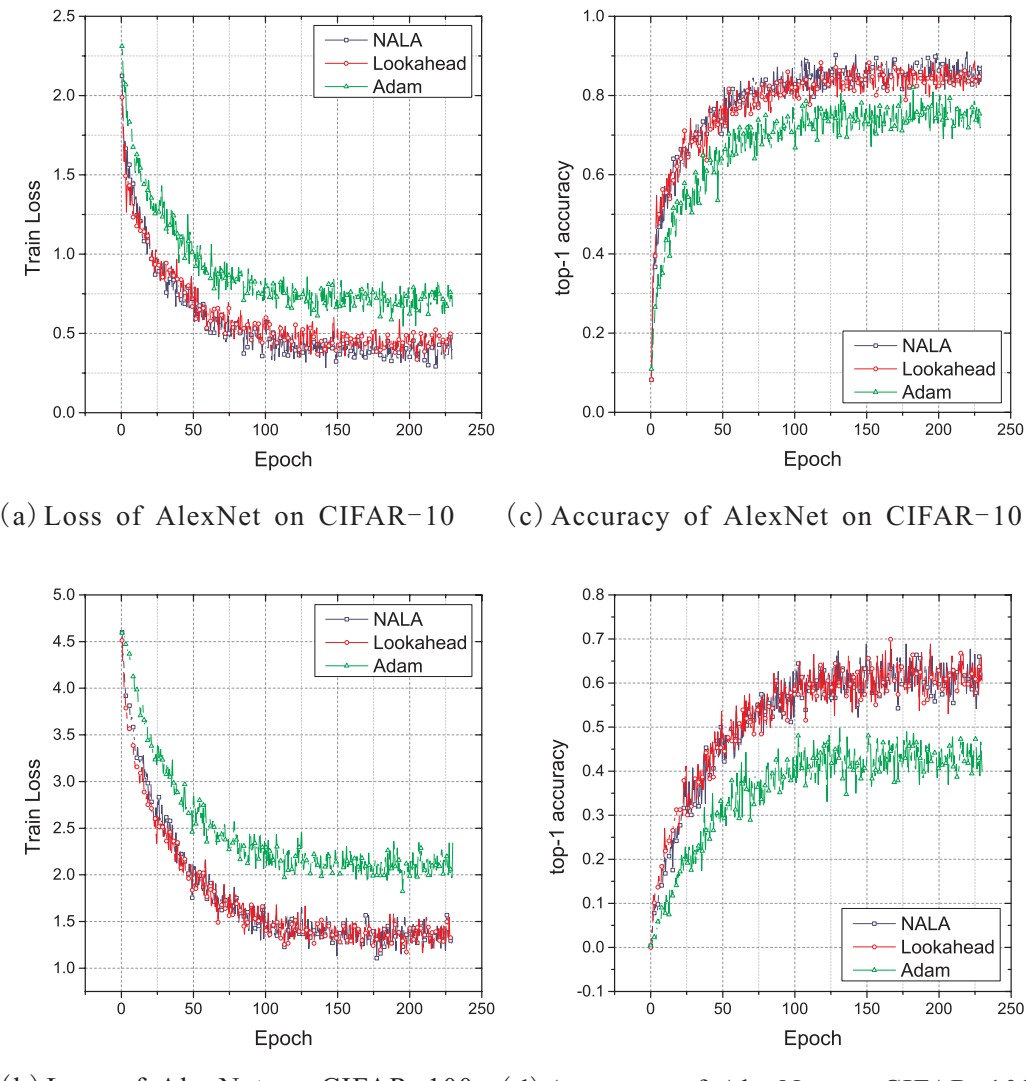

(a) Loss of AlexNet on CIFAR–10   (c) Accuracy of AlexNet on CIFAR–10

(b) Loss of AlexNet on CIFAR–100   (d) Accuracy of AlexNet on CIFAR–100

**Figure 3** (A–D) Train loss and top-1 accuracy of AlexNet trained by three different optimizers on CIFAR-10 and CIFAR-100.

layers respectively. The 18-layer ResNet (ResNet-18), which is considered to have a faster convergence speed (*He et al., 2015*), is applied for the CIFAR-10 and Fashion-MNIST experiments in this work. Three optimization algorithms, NALA, Lookahead, and Adam, are used for training the ResNet-18 model.

The standard cross-entropy objective function is used for these experiments on ResNet-18. The initial learning rate is set to 0.0002 for both NALA, Lookahead, and Adam. For NALA and Lookahead, the synchronization period $k$ is set to 5. Training curves of these experiments are shown in Figs. 4A and 4C for CIFAR-10, Figs. 4B and 4D for Fashion-MNIST. Table 5 shows the loss value and the top-1 accuracy rate of the ResNet-18 models trained with the three optimizers on the CIFAR-10 and Fashion-MNIST datasets.

**Table 3 The best records of AlexNet train loss and top-1 accuracy on CIFAR-10 and CIFAR-100 during the 230 epochs training.**

| AlexNet | CIFAR-10 | | CIFAR-100 | |
|---|---|---|---|---|
| | Loss value | Top-1 accuracy | Loss value | Top-1 accuracy |
| NALA | 0.265425265 | 0.91796875 | 1.098710179 | 0.69531250 |
| Lookahead | 0.324133724 | 0.89453125 | 1.132630944 | 0.69921875 |
| Adam | 0.527983427 | 0.82031250 | 1.887291193 | 0.50390625 |

**Table 4 The number of timesteps these optimization algorithms require to achieve 60% top-1 accuracy and 80% or 50% top-1 accuracy during the 230 epochs training.**

| AlexNet | CIFAR-10 | | CIFAR-100 | |
|---|---|---|---|---|
| | 60% accuracy | 80% accuracy | 50% accuracy | 60% accuracy |
| NALA | 2,800 | 9,900 | 9,500 | 17,000 |
| Lookahead | 2,400 | 10,500 | 9,600 | 16,700 |
| Adam | 6,000 | 36,600 | 25,600 | – |

Table 6 gives the number of timesteps the three optimization algorithms require to achieve 70% and 90% top-1 accuracy during training.

As shown in Fig. 4 and Table 5, the ResNet-18 models trained with NALA and Adam exhibit almost the same performance by achieving very close loss values and the same top-1 accuracy on both CIFAR-10 and Fashion-MNIST datasets. The two algorithms have a significant advantage over Lookahead not only in accuracy but also in learning speed; see Table 6. Although Lookahead also applies the parameter averaging method to update its outer loop weights, its exploration trajectories may converge to suboptimal weights on ResNet models. The Nesterov momentum used by NALA may lead to a faster convergence direction for the optimization.

In general, NALA exhibits superior or comparable performance to the other popular optimization algorithms for the image classification tasks on the CIFAR-10, CIFAR-100 and Fashion-MNIST datasets, except when training AlexNet on CIFAR-100, where Lookahead achieves a slightly higher accuracy rate. The results of these experiments reveal the exceptional ability of employing the Nesterov accelerated gradient and the exponential moving average of weights in inner and outer loops to enhance deep learning. Our experiments demonstrate that NALA can effectively solve practical deep learning problems on the classical CNN models and public image datasets.

## ROBUSTNESS TO THE HYPERPARAMETERS

The hyperparameters of NALA are searched over to find good settings with which the algorithm can achieve satisfied optimization performance on the image classification tasks. Interestingly, the results show the robustness of NALA to its hyperparameters (*i.e.,* the synchronization period $k$, the step size of slow weights $\alpha$, and the decay factor $\mu$). This work evaluates the algorithm robustness to its hyperparameters by implementing NALA with varied settings of $k, \alpha, \mu$ and an initial learning rate of 0.001 for Adam optimizer in

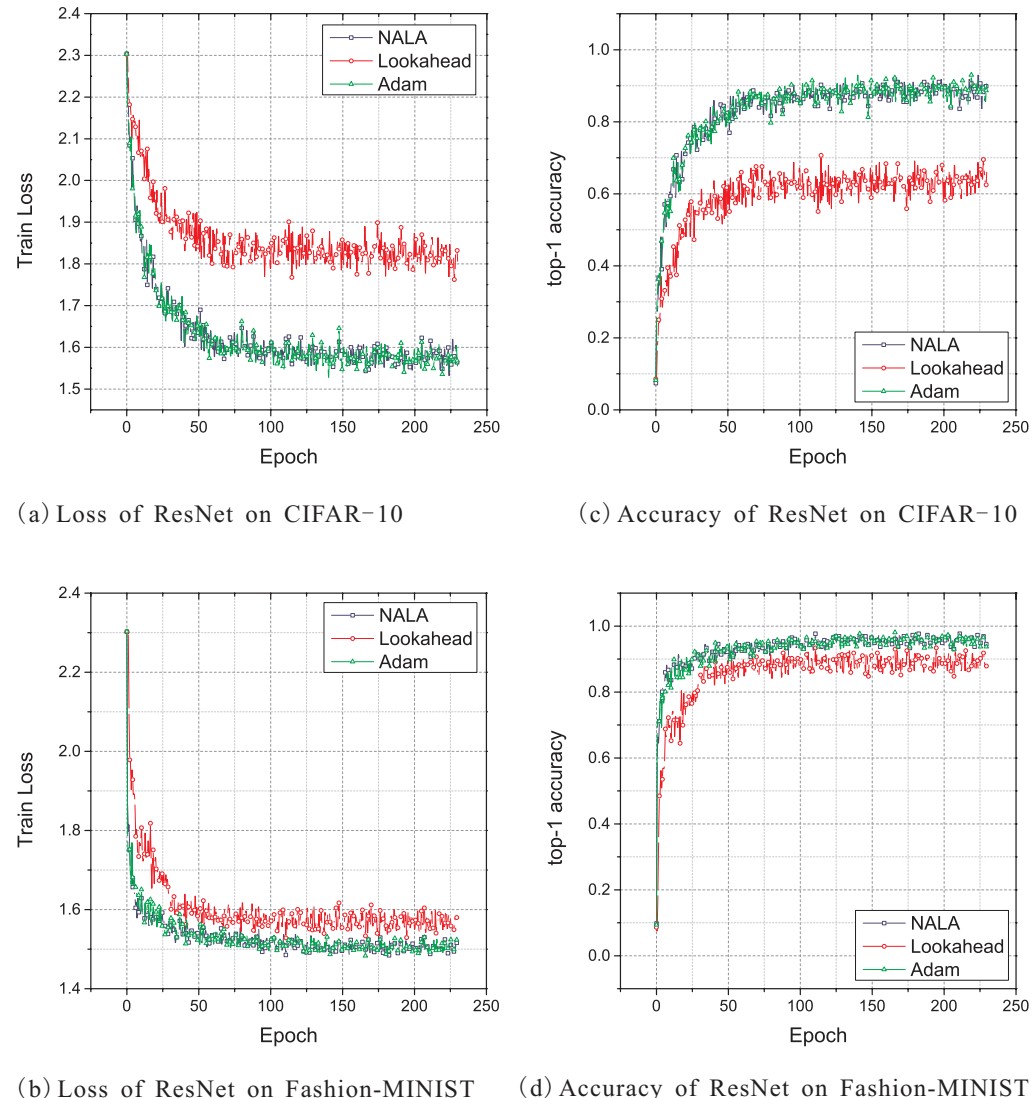

(a) Loss of ResNet on CIFAR−10

(c) Accuracy of ResNet on CIFAR−10

(b) Loss of ResNet on Fashion-MINIST

(d) Accuracy of ResNet on Fashion-MINIST

**Figure 4** **(A–D) Train loss and top-1 accuracy of ResNet-18 trained by three different optimizers on CIFAR-10 and Fashion-MINIST.**

the inner loop. The classification tasks on CIFAR-10, CIFAR-100, and Fashion-MNIST are performed to observe the optimization with different settings of the three hyperparameters.

The experimental results show that, NALA is robust to a wide range of hyperparameter settings, as shown in Tables 7, 8 and 9. For the image classification tasks involving different models and datasets, NALA consistently achieves fast convergence and acceptable accuracy across different settings of the hyperparameters, including the synchronization period $k$, the step size of slow weights $\alpha$ and the decay factor $\mu$. These experiments demonstrate that NALA is less sensitive to suboptimal hyperparameters, thereby reducing the need for extensive hyperparameter tuning.

**Table 5** The records of ResNet-18 train loss and top-1 accuracy on CIFAR-10 and Fashion-MNIST during the 230 epochs training.

| ResNet-18 | CIFAR-10 | | Fashion-MNIST | |
|---|---|---|---|---|
| | Loss value | Top-1 accuracy | Loss value | Top-1 accuracy |
| NALA | 1.525221467 | 0.93750000 | 1.479115725 | 0.98046875 |
| Lookahead | 1.762263298 | 0.70703125 | 1.526486874 | 0.93359375 |
| Adam | 1.521514177 | 0.93750000 | 1.48304069 | 0.98046875 |

**Table 6** The number of timesteps these optimization algorithms require to achieve 70% and 90% top-1 accuracy during the 230 epochs training.

| ResNet-18 | CIFAR-10 | | Fashion-MNIST | |
|---|---|---|---|---|
| | 70% accuracy | 90% accuracy | 70% accuracy | 90% accuracy |
| NALA | 2,800 | 18,700 | 400 | 3,000 |
| Lookahead | 22,400 | – | 1,500 | 9,700 |
| Adam | 3,900 | 17,500 | 300 | 2,300 |

**Table 7** The records of train loss and top-1 accuracy during training the classification models with $\mu = -0.5$ and $\alpha = 0.001$ across different synchronization period $k$ settings.

| Period $k$ | LeNet-5 & CIFAR-100 | | AlexNet & CIFAR-10 | | ResNet-18 & Fashion-MNIST | |
|---|---|---|---|---|---|---|
| | Loss value | Top-1 accuracy | Loss value | Top-1 accuracy | Loss value | Top-1 accuracy |
| 3 | 0.891243339 | 0.73828125 | 0.257579595 | 0.91406250 | 1.469873652 | 0.98106825 |
| 4 | 1.072080374 | 0.71484375 | 0.270872831 | 0.91015625 | 1.501279878 | 0.97956500 |
| 5 | 0.958603263 | 0.72656250 | 0.265425265 | 0.91796875 | 1.479115725 | 0.98046875 |
| 6 | 1.004980326 | 0.73046875 | 0.307197988 | 0.91406250 | 1.616237893 | 0.97228245 |
| 7 | 0.925782800 | 0.73437500 | 0.330697894 | 0.89843750 | 1.653237660 | 0.96428785 |

## CONCLUSION

This article presents NALA, an optimization algorithm combining NAG with the Adam optimizer. NALA adopts a modified look-ahead scheme with parameter averaging to derive its extrapolation point for computing the accelerated gradient and Nesterov momentum. Although NALA has a marginal improvement over Lookahead, it updates parameters along the direction of Nesterov accelerated gradient instead of only by parameter averaging as in Lookahead. That makes the algorithm see a slight future on the loss surface, so as to avoid missing the global optimum. Additionally, NALA only requires first-order gradients with minimal memory and computation overhead. The experimental results show that NALA works well in practice and compares favorably to other popular optimizers, regardless of different hyperparameter settings.

Future work could aim to test different inner loop optimizers and find a more efficient one to be combined with the modified look-ahead scheme. Our NALA algorithm integrates the standard Adam optimizer, which is one of the most widely used optimizers in deep learning, into inner loops in order to take advantage of its adaptive learning rate. Current optimization algorithms based on Adam (*e.g.*, RAdam (*Liu et al., 2020*), Adan

**Table 8** The records of train loss and top-1 accuracy during training the classification models with $k = 5$ and $\alpha = 0.001$ across different decay factor $\mu$ settings.

| Decay factor $\mu$ | LeNet-5 & CIFAR-100 | | AlexNet & CIFAR-10 | | ResNet-18 & Fashion-MNIST | |
|---|---|---|---|---|---|---|
| | Loss value | Top-1 accuracy | Loss value | Top-1 accuracy | Loss value | Top-1 accuracy |
| −0.3 | 0.951563597 | 0.73437500 | 0.322865337 | 0.88281250 | 1.683645686 | 0.95457450 |
| −0.4 | 0.954275727 | 0.75000000 | 0.358601660 | 0.88671875 | 1.699346565 | 0.94953465 |
| −0.5 | 0.958603263 | 0.72656250 | 0.265425265 | 0.91796875 | 1.479115725 | 0.98046875 |
| −0.6 | 0.961694419 | 0.73828125 | 0.277541548 | 0.91015625 | 1.530345548 | 0.97365450 |
| −0.7 | 0.912176311 | 0.72656250 | 0.276719004 | 0.91406250 | 1.525757385 | 0.97457385 |

**Table 9** The records of train loss and top-1 accuracy during training the classification models with $k = 5$ and $\mu = -0.5$ across different step size $\alpha$ settings.

| Step size $\alpha$ | LeNet-5 & CIFAR-100 | | AlexNet & CIFAR-10 | | ResNet-18 & Fashion-MNIST | |
|---|---|---|---|---|---|---|
| | Loss value | Top-1 accuracy | Loss value | Top-1 accuracy | Loss value | Top-1 accuracy |
| 0.0001 | 0.982763350 | 0.74609375 | 0.278902113 | 0.91015625 | 1.514573346 | 0.97834645 |
| 0.001 | 0.961694419 | 0.73828125 | 0.265425265 | 0.91796875 | 1.479115725 | 0.98046875 |
| 0.01 | 1.003419161 | 0.71875000 | 0.317181528 | 0.90625000 | 1.593457834 | 0.96557850 |

(*Xie et al., 2022*), and AdaXod (*Liu & Li, 2023*)) have achieved numerous advancements in the field of machine learning. We believe that the combination of state-of-the-art adaptive optimizers and our Nesterov accelerated look-ahead scheme could be meaningful work for improving optimization algorithms. We leave this work to future research.

## ACKNOWLEDGEMENTS

The authors would like to thank Professor Zhun-Ga Liu for his helpful discussions and guidance.

### Funding

The authors received no funding for this work.

### Competing Interests

The authors declare there are no competing interests.

### Author Contributions

- Xuan Zuo conceived and designed the experiments, performed the experiments, analyzed the data, performed the computation work, prepared figures and/or tables, authored or reviewed drafts of the article, and approved the final draft.
- Hui-Yan Li analyzed the data, authored or reviewed drafts of the article, and approved the final draft.

- Shan Gao performed the experiments, prepared figures and/or tables, and approved the final draft.
- Pu Zhang conceived and designed the experiments, authored or reviewed drafts of the article, and approved the final draft.
- Wan-Ru Du performed the computation work, prepared figures and/or tables, and approved the final draft.

## Data Availability

The code for the NALA optimizer and the classification experiments is available at GitHub and Zenodo:

- https://github.com/guanzhongzx/Nesterov-Look-ahead

- xuanzuo. (2024). guanzhongzx/Nesterov-Look-ahead: NALA optimizer algorithm (v1.0). Zenodo. https://doi.org/10.5281/zenodo.11296853

The experimental data of LeNet-5, AlexNet and ResNet-18 trained by NALA is available at GitHub and Zenodo:

- https://github.com/guanzhongzx/OptimizerPerformance

- xuanzuo. (2024). guanzhongzx/OptimizerPerformance: NALA experimental results v1.0 (v1.0). Zenodo. https://doi.org/10.5281/zenodo.11296354.

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
