# Peer review of "NALA: a Nesterov accelerated look-ahead optimizer for deep learning"

_PeerJ Computer Science, doi:10.7717/peerj-cs.2167_

## Round 0.1 · original submission · Major Revisions

Both reviewers find some merit in the work presented, but also identify a number of issues for improvement.

**Language Note:** PeerJ staff have identified that the English language needs to be improved. When you prepare your next revision, please either (i) have a colleague who is proficient in English and familiar with the subject matter review your manuscript, or (ii) contact a professional editing service to review your manuscript. PeerJ can provide language editing services - you can contact us at [email protected] for pricing (be sure to provide your manuscript number and title). – PeerJ Staff

Reviewer 1 ·

Basic reporting

The article introduces a modified look-ahead scheme including Nestorov momentum into the Lookahead algorithm. The paper is well written, the english is good. The paper is divided in 6 parts.
1. The introduction is clear, well written. We understand the purpose of the research and how the paper is organised.
2. The background part is well documented, many references are cited to give the state of the art in deep learning optimizer methods.
3. Part 3 describes the Nestorov accelerated Lookahead Algorithm. This part needs to be rewritten. The authors should explain every notation they use in the equations, even if they apply the usual formalism. What is 'y', 'Mu' 'Teta' 'alpha' 'J' ...? This remark applies to equations (1) and (2), but also to Algorithm 1 and equations 3, 4 and 5. The authors should also evaluate the calculation time impact at this point from a theoretical point of view.
4 Part 4 discusses the experiments conducted on image classification datasets on which two CNN models (LeNet-5 and AlexNet) are compared. The authors should justify their choices (number of epochs, different parameters used to perform the learning process of the different methods). The results show a good convergence of NALA, a "faster learning speed" but the calculation time gained or lost is not mentioned or compared. This could be problematic if time was lost because of a more complex calculation. Figures should be improved, the superposition of colors makes the figure interpretation difficult.
5. Checking the robustness of the hyperparameters is very useful. It would have been interesting to check the robustness of the method on another dataset as well.
6. In the conclusion, the authors could talk about their future work and perspectives.

Experimental design

no comment

Validity of the findings

no comment

Cite this review as

Reviewer 2 ·

Basic reporting

The authors propose an optimization algorithm which combines Nesterov's Accelerated Gradient (NAG) algorithm with the Adam optimizer, which they call Nesterov Accelerated Look-Ahead (NALA) algorithm. They demonstrate the algorithm on two classical convolutional neural networks, LeNet-5 and AlexNet, and two classical image classification datasets, CIFAR-10 and CIFAR-100.
The following observations should be addressed:
1. The notations in the equations should be thoroughly explained, even if they are similar to the ones used in the literature, in order to improve the readability of the paper.

Experimental design

2. Only two very simple convolutional neural network architectures are used to demonstrate the proposed algorithm. More complex architectures, like ResNet, for example, should also be used.
3. The datasets used for the experiments, CIFAR-10 and CIFAR-100, are considered nowadays too small datasets. Bigger datasets, like ImageNet, for example, should also be used.

Validity of the findings

4. The results reported in the tables are not the most relevant, as the authors could train all the algorithms to convergence, and report the results then, because only in that way are we able to see the faster convergence of some algorithms as opposed to others.

Additional comments

5. Minor points: “high-dimensional parameters spaces” should be “high-dimensional parameter spaces”, “the training of DNN” should be “the training of DNNs”, “convergence, furthermore” should be “convergence. Furthermore”, “In section 4” should be “In Section 4”, “which combines the advantages” should be “which combine the advantages”, “of different neural network” should be “of different neural networks”, “then compute some” should be “then computes some”, “can combines parameter averaging with any standard optimizers” should be “can combine parameter averaging with any standard optimizer”, “Equation 1” should be “Equation (1)”, “colour” should be “color”, “with above” should be “with the above”, “to avoid the missing” should be “to avoid missing”.
6. In general, the article should be reviewed to correct spelling errors, and to improve the English language.

Cite this review as

---

## Round 0.2 · accepted · Accept

Authors revised their paper according to the comments of the reviewers. It can be accepted now.

Reviewer 2 ·

Basic reporting

no comment

Experimental design

no comment

Validity of the findings

no comment

Additional comments

The authors have successfully addressed the majority of the issues raised by the reviewers. The paper can be accepted in its current form.

Cite this review as